# OPEN SET FACE FORGERY DETECTION VIA DUAL-LEVEL EVIDENCE COLLECTION

## ABSTRACT

The proliferation of face forgeries has increasingly undermined confidence in the authenticity of online content. Given the rapid development of face forgery generation algorithms, new fake categories are likely to keep appearing, posing a major challenge to existing face forgery detection methods. Despite recent advances in face forgery detection, existing methods are typically limited to binary Real-vs-Fake classification or the identification of known fake categories, and are incapable of detecting the emergence of novel types of forgeries. In this work, we study the *Open Set Face Forgery Detection (OSFFD)* problem, which demands that the detection model recognize novel fake categories. We reformulate the OSFFD problem and address it through uncertainty estimation, enhancing its applicability to real-world scenarios. Specifically, we propose the Dual-Level Evidential face forgery Detection (DLED) approach, which collects and fuses category-specific evidence on the spatial and frequency levels to estimate prediction uncertainty. Extensive evaluations conducted across diverse experimental settings demonstrate that the proposed DLED method achieves state-of-the-art performance, outperforming various baseline models by an average of 20% in detecting forgeries from novel fake categories. Moreover, on the traditional Real-versus-Fake face forgery detection task, our DLED method concurrently exhibits competitive performance.

## 1 INTRODUCTION

Deepfakes, which use deep learning techniques to generate or modify faces and voices, continue to rapidly increase in both sophistication and accessibility. The diversity of deepfake forgeries (Korshunova et al., 2017; Karras, 2017; Shen & Liu, 2017; Siarohin et al., 2019) causes different visual artifacts to appear in the generated deepfakes, making deepfake detection increasingly difficult. According to a survey by Mirsky et al. (Mirsky & Lee, 2021), existing face deepfake forgeries can generally be organized into four categories: Face Swapping (FS), Face Reenactment (FR), Entire Face Synthesis (EFS), and Face Editing (FE). As new generation methods continue to emerge, it is likely that novel categories of facial deepfakes will be developed.

Despite progress in deepfake detection under closed set scenarios (Yan et al., 2023b; Qian et al., 2020; Gu et al., 2022; Ni et al., 2022), where both training and testing data contain the same known fake forgeries[1], these methods have yet to fully address the challenge of generalizing to unseen fake forgeries. Some studies (Wang et al., 2020; Cao et al., 2022; Nadimpalli & Rattani, 2022; Zhuang et al., 2022; Sun et al., 2023) have proposed mechanisms to improve generalization to unseen forgeries. However, their overall performance remains suboptimal, and they fail to detect the emergence of novel fake categories.

In this paper, we study the Open Set Face Forgery Detection (OSFFD) problem to address this issue. OSFFD was proposed in (Diniz & Rocha, 2024; Zhou et al., 2024), but it remains an underexplored problem. Traditional deepfake detection and attribution tasks either distinguish between real and fake images or assign forgeries to predefined categories. In contrast, OSFFD determines

---

[1]In this paper, we define "fake forgeries" as specific deepfake methodologies, and "fake categories" as the broader groups to which these methodologies belong; e.g., FSGAN (Nirkin et al., 2019) is the fake forgery and Face Swapping is its according fake category.

whether a given face belongs to a novel fake category, while simultaneously performing multiclass classification among real and known fake categories. The difference among these settings is shown in Figure 1. The aforementioned studies approached the OSFFD problem by training models on labeled data for seen classes (real and known fake categories), and unlabeled data for novel fake categories. This setup has practical limitations as data from a novel fake category would not be integrated into datasets immediately after its proliferation. In this paper, we reformulate the OSFFD problem by restricting model training to only real and known fake categories, which enhances the real-world applicability of OSFFD.

To address the OSFFD problem, we formulate it as an uncertainty estimation issue that assesses the confidence of model predictions based on the evidence collected from the data. During training, the model is exposed to known fake categories and learns to assign low uncertainty to these samples. At test time, samples from unknown categories are expected to yield high uncertainty scores, facilitating their detection.

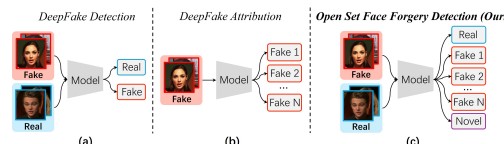

Figure 1: **Comparison with existing settings**. Different from DeepFake Detection (a) and Attribution (b), Open Set Face Forgery Detection (c) aims to identify whether a forgery originates from a novel fake category or not while simultaneously performing multiclass classification among real and known fake categories.

In this paper, we propose a novel *Dual-Level Evidential face forgery Detection* approach, DLED, that simultaneously identifies emerging, unknown fake categories and performs multiclass classification among real and known fake categories. To enable novel category recognition, DLED leverages Evidential Deep Learning (EDL) (Sensoy et al., 2018; 2020; Shi et al., 2020) for classification and uncertainty estimation. However, unlike conventional open set classification, OSFFD operates on structured facial imagery whose spatial semantic patterns alone are insufficiently discriminative (Wang et al., 2020). Accordingly, DLED augments these cues with complementary low-level frequency artifacts, yielding a more effective application of EDL. Because both sources are informative, detection decisions should reflect their joint support. To this end, we introduce an uncertainty-guided evidence fusion mechanism grounded in Dempster's combination rule (Sentz & Ferson, 2002), enabling DLED to integrate evidence on both the spatial and frequency levels into a unified, comprehensive uncertainty estimate. Furthermore, we propose an improved uncertainty estimation approach to enhance the model's capability to detect novel fake, as the original EDL formulation can be affected by evidence from irrelevant classes, resulting in suboptimal uncertainty quantification.

Compared with existing face forgery detection methods, a key advantage of our DLED model lies in its ability to promptly detect newly emerging fake categories and avoid misclassification, without relying on any prior knowledge of these categories. While existing deepfake detection algorithms can be adapted to be feasible in the OSFFD problem, e.g., one-class detectors (Shiohara & Yamasaki, 2022; Khalid & Woo, 2020; Larue et al., 2023) can combine with a separate multiclass classifier, they often struggle to balance between accurate novel category detection and effective multiclass classification. In addition, our methodology is grounded in principled reasoning, offering clear interpretability for the OSFFD results.

In summary, our contribution is three-fold:

- We reformulate the Open Set Face Forgery Detection (OSFFD) problem, eliminating the reliance on unlabeled data from novel fake categories during model training, making it more applicable in real life.

- We propose leveraging EDL to treat the OSFFD as an uncertainty estimation problem, enabling the model to determine whether a face image originates from a novel fake category.

- We propose the DLED approach, which aggregates and fuses evidence on both the spatial and frequency levels to estimate prediction uncertainty. Extensive empirical results validate its effectiveness and demonstrate its superiority over various baseline models.

## 2 RELATED WORK

**Deepfake Detection.** A wide range of deepfake detection approaches have been studied in the literature (Huang et al., 2022). Most existing methods (Sun et al., 2022; Ni et al., 2022; Zhuang

et al., 2022; Cao et al., 2022) leverage spatial patterns to detect manipulation artifacts, while others (Luo et al., 2021; Gu et al., 2022; Zhang et al., 2019) exploit discrepancies in the frequency domain to reveal forgery traces. Some studies also (Tan et al., 2024; Wang et al., 2023b; Guillaro et al., 2023) integrated features from complementary modalities, such as noise patterns, to further distinguish fake faces. One-class anomaly detection methods (Khalid & Woo, 2020; Shiohara & Yamasaki, 2022; Larue et al., 2023) treat real faces as the positive class and all other data as anomalous outliers, training the model exclusively on the positive class to distinguish between real and fake faces. Recent works (Ojha et al., 2023; Khan & Dang-Nguyen, 2024) find that the pretrained CLIP (Radford et al., 2021) model performs well on unseen forgeries. Based on this finding, several recent works (Liu et al., 2024b;a; Yang et al., 2025) designed diverse adaptations for CLIP to enhance its detection capabilities. However, these approaches are limited by their exclusive focus on Real-vs-Fake classification, which overlooks the differences among different fake categories.

**Deepfake Attribution.** The deepfake attribution task aims to identify the source of fake faces so that models can provide persuasive explanations for the results of deepfake detection. However, most of these methods (Wu et al., 2024; Huang et al., 2023; Yang et al., 2022; Zhong et al., 2023) are limited to the closed set scenario. Few methods have utilized the "open world" setting to track unseen forgeries. The open-world GAN (Girish et al., 2021) method is designed to detect images generated by previously unseen GANs, but its framework does not extend to other manipulations such as face editing. Another work, CPL (Sun et al., 2023), introduced a benchmark which encompasses a broader array of unseen forgeries derived from multiple known categories. However, this setting relies on access to unlabeled data from such forgeries during training and does not determine whether a given forgery originates from a novel category, thereby limiting its practical applicability. Although recent works (Wang et al., 2024a; 2023a) introduced open set classification for forgeries, their settings do not differentiate between unseen forgeries originating from known categories and those from entirely novel categories, nor can they determine whether a face is real or fake.

**Open Set Recognition.** Open Set Recognition is a well-defined task that recognizes known classes and differentiates the unknown. The pioneering work (Scheirer et al., 2012) formalized the definition and introduced a "one-vs-set" machine based on binary SVM. Prototype learning and metric learning methods (Chen et al., 2021; Yang et al., 2020; Zhang & Ding, 2021) have been applied to identify the unknown by keeping unknown samples at large distances to prototypes of known class data. Recently, uncertainty estimation methods (Wang et al., 2021; Bao et al., 2021; Fan et al., 2024; 2023) using Evidential Deep Learning (EDL) have shown promising results on open set recognition problems. EDL (Sensoy et al., 2018; 2020; Shi et al., 2020) works well to quantify model confidence and prediction uncertainty, exhibiting high efficacy in handling unseen data types, and it has been further broadened to encompass multi-view classification (Han et al., 2020; Huang et al., 2024). To the best of our knowledge, this paper is the first to integrate EDL into the OSFFD problem.

# 3 OPEN SET FACE FORGERY DETECTION

**Definition.**
As depicted in Figure 2, Open Set Face Forgery Detection (OSFFD) addresses a practical problem: leveraging knowledge from seen classes (i.e., real faces and faces from known fake categories) to classify a given face as either belonging to a seen class or to the newly emerging, unseen fake category. In the training phase, the model is exposed exclusively to images from seen classes, while images from novel fake categories are reserved for testing purposes.

**Motivation.** OSFFD requires a model to simultaneously discover novel fake categories and perform multiclass classification. Among these two objectives, novel fake category discovery is the core challenge. However, most existing detectors (Ojha et al., 2023; Yan et al., 2024a) emphasize out-of-distribution (OOD) generalization, which target binary real-vs-fake discrimination on unseen testing

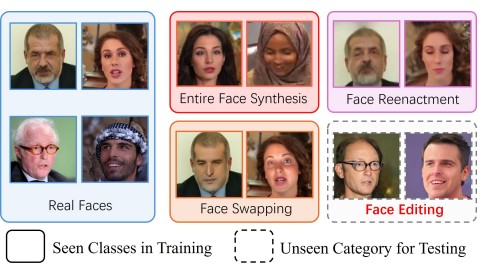

Figure 2: **Illustration for Fake Categories in OSFFD.** Real faces and fake faces from the seen categories are used to train the model. Subsequently, the model is evaluated on test data that includes both seen classes and previously unseen categories. In the figure, the labels EFS, FR, and FS denote seen categories, whereas FE represents an unseen category.

samples. As a result, they neither support multiclass classification nor distinguish novel fake categories, rendering them unsuitable for OSFFD. One alternative is a two-stage pipeline that first partitions samples into seen versus unseen class via OOD detection (Khalid & Woo, 2020; Shiohara & Yamasaki, 2022) and then applies a face-forgery classifier to seen classes; but this decoupled design optimizes different training objectives across stages and offers limited theoretical interpretability. Additionally, existing open set recognition (OSR) methods (Zhang & Xiang, 2023; Lang et al., 2024) could hardly perform well when directly applied to OSFFD as the data in OSFFD consist of highly structured facial imagery that requires additional mechanisms to extract discriminative representations. Therefore, novel algorithms need to be developed to address the OSFFD problem.

**Formulation.** Given a labeled training set $D_S = \{(x_i, y_i)\}_{i=1}^{M}$ consisting of $M$ labeled samples from $K$ seen classes comprising the Real class and $N$ known fake categories (i.e., $K = N + 1$, $y_i \in \{1, \ldots, K\}$) and a test set $D_T$ containing samples from the face class set $\{R, F_1, \ldots, F_N, F_{N+1}, \ldots, F_{N+U}\}$, where $U$ is the number of unknown fake categories, we denote the embedding space of class $k \in [1, K]$ as $P_k$, and its corresponding open space as $O_k$. The open space is further divided into two subspaces: the positive open space from other known classes $O_k^{\text{pos}}$ and the negative open space $O_k^{\text{neg}}$ that represents the remaining infinite unknown region.

For a single class $k$, the samples $D_S^k \in P_k$, $D_S^{\neq k} \in O_k^{\text{pos}}$, and $D_V \in O_k^{\text{neg}}$ are positive training data, negative training data and potential unknown data respectively. Then, we could use a simple binary classification model $\Psi_k(x) \to \{0, 1\}$ to detect unseen classes (Chen et al., 2021) and optimize the model by minimizing the expected risk $R^k$:

$$\underset{\Psi_k}{\arg\min} R^k = R_c(\Psi_k, P_k \cup O_k^{\text{pos}}) + \alpha \cdot R_o(\Psi_k, O_k^{\text{neg}}), \tag{1}$$

where $\alpha$ is a positive constant, $R_c$ is the empirical classification risk on the known data, and $R_o$ is the open space risk (Scheirer et al., 2012). $R_o$ measures the likelihood of labeling unknown samples as either known or unknown classes, expressed as a nonzero integral function over the space $O_k^{\text{neg}}$ :

$$R_o(\Psi_k, O_k^{\text{neg}}) = \frac{\int_{O_k^{\text{neg}}} \Psi_k(x) dx}{\int_{P_k \cup O_k} \Psi_k(x) dx}. \tag{2}$$

The more frequently the negative open space $O_k^{\text{neg}}$ is labeled as positive, the higher the associated open space risk.

We extend single-class detection to the multiclass OSFFD setting by integrating multiple binary classification models $\Psi_k$ using a one-vs-rest strategy. With Eq. 1, the overall expected risk is computed as the sum over all seen classes: $\sum_{k=1}^{K} R^k$. This is equivalent to training a multiclass classification model $\mathcal{F} = \odot(\Psi_1, \ldots, \Psi_K)$ for $K$-class classification, where $\odot(\cdot)$ denotes the integration operation. The overall training optimization objective is formulated as:

$$\arg\min\{R_c(\mathcal{F}, D_S) + \alpha \cdot \sum_{k=1}^{K} R_o(\mathcal{F}, D_V)\}, \tag{3}$$

which demands the model to minimize the combination of the classification risk on seen classes and the open space risk on unseen classes. Therefore, our goal is to train a multiclass classification model $\mathcal{F}(\cdot)$, parameterized by $\theta$, on $K$ seen classes to accurately classify faces as either real or belonging to one of the known fake categories, while simultaneously detecting novel fake categories as a distinct $(K + 1)^{\text{th}}$ class. We further formulate OSFFD as an uncertainty estimation problem: the model $\mathcal{F} : \mathcal{X} \to (\tilde{y}, \tilde{u})$ outputs a predicted category label $\tilde{y} \in \{1, \ldots, K\}$ and its associated predictive uncertainty $\tilde{u}$. If the predictive uncertainty exceeds the class-specific threshold $\tau_{\tilde{y}}$, i.e., $\tilde{u} > \tau_{\tilde{y}}$, the predicted label is deemed unreliable and the instance is assigned to the novel fake category.

## 4 METHODOLOGY

To solve the formulated uncertainty estimation problem, we utilize established techniques such as MaxLogit and Evidential Deep Learning.

**Plug-in OSR Techniques.** Maximum Softmax Probability (Hendrycks et al., 2019) and MaxLogit (Wang et al., 2022) detectors are two widely used plug-in OSR techniques, which utilize the maximum Softmax probabilities and the maximum logits as the model prediction confidence with no extra computational costs.

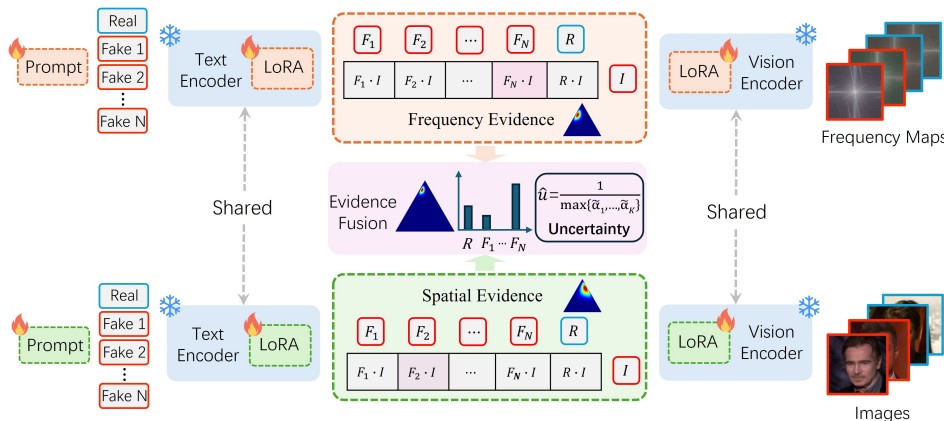

Figure 3: **Overview of DLED.** DLED collects and fuses evidence from both the spatial and frequency domains to estimate prediction uncertainty. Our improved uncertainty estimation $\hat{u}$ is applied to achieve better detection performance. $F_N$ represents the $N$-th fake category and $K$ is the total known class number. If the uncertainty for the given sample is larger than the computed threshold, its label will be reassigned to the novel fake category. In the evidence illustration, we present a demonstration of a three-class classification scenario ($K = 3$).

**Evidential Deep Learning.** Evidential Deep Learning (EDL) is an effective technique that performs multiclass classification and uncertainty modeling by introducing the framework of Dempster-Shafer Theory (Sentz & Ferson, 2002) and subjective logic (Jøsang, 2016). For a $K$-class classification problem, given a sample $x$ and a model $\mathcal{F}$ parameterized by $\theta$, the predicted evidence is given by $e = h(\mathcal{F}(x;\theta)) \in R^K$, where $h$ is an evidence function. With total strength $S = \sum_{k=1}^{K} \alpha_k$, where $\alpha_k = e_k + 1$, the predicted probability for class $k$ is $p_k = \alpha_k/S$ and the prediction uncertainty $u$ is calculated as $u = K/S$. EDL has been useful to detect data from unknown classes in prior literature (Bao et al., 2021; Zhao et al., 2023; Yu et al., 2024; Wang et al., 2024b; Peng et al., 2025). These literature motivates us to develop a EDL-based algorithm to detect novel deepfake categories. Compared with plug-in OSR techniques, EDL provides a more principled uncertainty estimation.

**Challenges in applying EDL.** In our approach, we employ EDL to collect evidence for face forgery detection. However, leveraging EDL to address the OSFFD problem meets the following challenges:

1) How to collect sufficient evidence in the OSFFD problem? Unlike conventional open set image classification, face forgery detection involves highly structured facial imagery. As a result, off-the-shelf EDL do not directly carry over to OSFFD with satisfactory performance. To bridge this gap, we extract evidential cues at two complementary levels: high-level semantic signals in the spatial domain and low-level artifacts in the frequency domain.

2) How can we achieve a comprehensive integration of collected complementary evidential cues? As both sources carry informative evidence, detection decisions should account for their joint contribution. The key challenge, therefore, is integrating the two independent uncertainty estimates into one well-calibrated and comprehensive metric. We address this issue by proposing a novel uncertainty-guided evidence fusion mechanism.

## 5 DUAL LEVEL EVIDENCE COLLECTION

**Overview.** To address the OSFFD problem, we propose the Dual-Level Evidential face forgery Detection (DLED) approach, which is exhibited in Figure 3. DLED exploits EDL through a dual-level evidential architecture that captures category characteristics of facial imagery across spatial and frequency domains, yielding sufficiently discriminative evidence. It addresses the evidence aggregation challenge with an uncertainty-guided fusion mechanism and further incorporates an uncertainty-improvement procedure to enhance the reliability of the resulting estimates. Together, these components enable DLED to detect novel fake categories by quantifying classification uncertainty across complementary levels and determining whether an existing prediction should be reassigned to the novel category.

**Spatial and Frequency Evidence.** Our DLED model addresses the evidence collection problem by extracting cues at two complementary levels: high-level spatial semantic signals and low-level frequency artifacts. Face forgeries generally fall into several common categories (FS, FR, EFS, and FE) based on their characteristics in the context of human visuals (Mirsky & Lee, 2021). We refer to these characteristics as deepfake category semantics, which is neglected by most existing works. Exploiting these semantics, the model can discern subtle differences among fake categories. To leverage both contextual and visual deepfake semantics, we employ the CLIP (Radford et al., 2021) architecture, a vision-language model designed to align image and text representations in a shared semantic space. Given an input image and the class textual descriptions, we then calculate the logit mass $m_i$ for class $i$. In contrast to standard open set classification, relying solely on visual semantics fails to capture the structure of forgery images. We thus leverage low-level artifacts in the frequency domain as a complementary source of evidence. Specifically, for each input image, we obtain its frequency map by applying the Fast Fourier Transform and shifting the resulting spectrum to center the low-frequency components, thereby making them more prominent. To extract evidence from these complementary domains, we employ two parallel CLIP pipelines, each with a dedicated image encoder and text encoder. Since CLIP is not explicitly trained to capture forgery image patterns, particularly in the frequency domain, we adapt it by fine-tuning the encoders along with the text prompts while freezing all other pretrained parameters. For text prompts, we employ Context Optimization (Zhou et al., 2022), which augments class tokens with learnable prompt vectors to yield stronger context embeddings. For image and text encoders, we integrate LoRA (Hu et al., 2022) layers into them, which enhance the model's understanding of deepfakes while not adding any additional parameters during testing. Although we have two parallel branches for spatial and frequency level representations, we reduce memory consumption by sharing their pretrained parameters.

**Evidential Uncertainty Estimation.** Our DLED model detects novel fake categories through evidential uncertainty estimation using Evidential Deep Learning (EDL) (Sensoy et al., 2018) in an end-to-end manner grounded in solid theoretical principles. EDL employs deep neural networks to output the parameters of a Dirichlet distribution over class probabilities, which is then used for both class prediction and uncertainty estimation. This process can be regarded as an evidence collection process. By leveraging EDL, our method quantifies the uncertainty associated with each prediction to assess its reliability. If the uncertainty is high, the model will reclassify the input as belonging to the novel class, thereby enabling the identification of faces from previously unseen fake categories.

Specifically, for each of the spatial and frequency branches with classification logits mass $m$, our approach calculates the corresponding evidence $e = h(m)$ using an evidence function $h(\cdot)$ that guarantees $e$ to be non-negative. During the training phase, to facilitate evidence collection, we independently apply the following EDL loss to each branch:

$$\mathcal{L}_{EDL}(e, y) = \sum_{k=1}^{K} y_k(\log S - \log(e_k + 1)), \tag{4}$$

where $S = \sum_k \alpha_k$ and $\alpha_k = e_k + 1$, denoting the total strength of the Dirichlet distribution governed by $\{\alpha_{1,...,K}\}$, and $y$ is the one-hot $K$-class label. We also apply AvU regularization (Bao et al., 2021; Hammam et al., 2022) to each branch for uncertainty calibration. The EDL loss and AvU regularization minimize $R_c$ and $R_o$ in Eq. 3 separately.

**Test-time Evidence Fusion.** To address the integration problem, we design a uncertainty-guided test-time evidence fusion mechanism. During model inference, according to EDL (Sensoy et al., 2018), the probabilities of different classes (belief masses) and the overall uncertainty mass can be calculated by $b_k = e_k/S$ and $u = K/S$. The $K$ belief mass values and the uncertainty $u$ are all non-negative and follow the sum-to-one rule: $\sum_{k=1}^{K} b_k + u = 1$. With this approach, we can get the belief and uncertainty for each branch.

Our DLED model collects two independent sets of probability mass $M^s = \{\{b_k^s\}_{k=1}^K, u^s\}$ and $M^f = \{\{b_k^f\}_{k=1}^K, u^f\}$ from the spatial and frequency domains. Inspired by previous works (Han et al., 2020; 2022), we apply the Dempster's combination rule (Sentz & Ferson, 2002) to get the joint detection probability mass set $\widetilde{M}$ in the following manner: $\widetilde{M} = M^s \oplus M^f$. The specific calculation rules for belief mass and uncertainty mass are formulated as

$$\widetilde{b}_k = \gamma(b_k^s b_k^f + b_k^s u^f + b_k^f u^s), \quad \widetilde{u} = \gamma u^s u^f, \tag{5}$$

where $\gamma = 1/(1 - \sum_{i \neq j} b_i^s b_j^f)$ is the scaling factor, normalizing the mass fusion to mitigate the effects of conflicting information between the spatial mass and frequency mass. With this newly obtained joint detection mass $\widetilde{M}$, the joint evidence and the parameters of the Dirichlet distribution are calculated as follows:

$$\widetilde{S} = \frac{K}{\widetilde{u}}, \;\; \widetilde{e}_k = \widetilde{b}_k \times \widetilde{S}, \;\; \text{and} \;\; \widetilde{\alpha}_k = \widetilde{e}_k + 1. \tag{6}$$

For a test sample $x^i$, the model prediction $\widetilde{p}_k^i$ for class $k$ is computed as $\widetilde{p}_k^i = \widetilde{\alpha}_k^i / \widetilde{S}^i$.

**Improved Uncertainty Estimation.** Considering $\widetilde{u} = K/\widetilde{S}$ and $\widetilde{S} = \sum_{k=1}^{K}(\widetilde{e}_k + 1)$, after dividing numerator and denominator by $K$, the uncertainty can be expressed as

$$\widetilde{u} = \frac{1}{1 + \frac{1}{K}\sum_1^K \{\widetilde{e}_{1,\dots,K}\}}, \tag{7}$$

which indicates that the uncertainty is assessed using the average evidence across all $K$ classes. Therefore, when the input data shows high evidence from irrelevant classes, the estimated uncertainty will be overestimated resulting in a sub-optimal estimation. To solve this problem, we propose an improved uncertainty estimation by replacing the *average* evidence with *maximum* evidence:

$$\hat{u} = \frac{1}{1 + \max\{\widetilde{e}_{1,\dots,K}\}} = \frac{1}{\max\{\widetilde{\alpha}_{1,\dots,K}\}} \tag{8}$$

where $\{\widetilde{e}_{1,\dots,K}\}$ represents the set of $K$ fused evidences. Our improved uncertainty measure offers the advantage of being less affected by low-evidence classes while retaining a normalized range between 0 and 1 for better human understanding. Moreover, it directly reflects the model's confidence in the predicted class. We recalculate the uncertainty $\hat{u}$ with Eq. 8 after the evidence fusion to get better detection performance.

To determine if a face image belongs to an unseen fake category, our model compares its uncertainty $\hat{u}$ with the uncertainty threshold for its predicted class. If the uncertainty falls above the threshold, the model reassigns the label to the novel category.

Table 1: Comparisons of model performance with diverse baseline methods implemented by ourselves for the OSFFD problem. We use different data configurations for the seen and unseen fake categories. For "FS", "FR", and "EFS", we let each fake category be the unseen category and let the left two be seen categories. For "FE & SM", we take FS, FR and EFS as seen categories and let FE and SM be the unseen categories. The best results are highlighted in **bold**.

| | Methods | FS | | FR | | EFS | | FE & SM | | Avg | |
|---|---|---|---|---|---|---|---|---|---|---|---|
| | | Acc | DR | Acc | DR | Acc | DR | Acc | DR | Acc | DR |
| Two-stage | OC-FakeDect (Khalid & Woo, 2020) | 58.16 | 14.68 | 60.69 | 11.43 | 56.14 | 9.01 | 56.74 | 11.67 | 57.93 | 11.70 |
| | SBI (Shiohara & Yamasaki, 2022) | 65.15 | 1.07 | 64.19 | 3.00 | 61.24 | 0.91 | 62.27 | 0.66 | 63.21 | 1.41 |
| CNN-based + OSR | Xception (Rossler et al., 2019) | 64.60 | 23.90 | 53.51 | 29.06 | 57.62 | 22.70 | 55.28 | 29.04 | 57.75 | 26.17 |
| | SPSL (Liu et al., 2021) | 65.07 | 16.71 | 54.10 | 18.93 | 59.67 | 18.12 | 60.02 | 25.98 | 59.71 | 19.93 |
| | SIA (Sun et al., 2022) | 62.09 | 13.59 | 54.62 | 13.36 | 56.85 | 10.99 | 56.29 | 22.53 | 57.46 | 15.12 |
| | UCF (Yan et al., 2023a) | 65.08 | 0.30 | 50.98 | 0.20 | 52.95 | 1.28 | 52.69 | 1.80 | 55.42 | 0.89 |
| | NPR (Tan et al., 2024) | **75.37** | 17.37 | 64.63 | 6.75 | 70.43 | 4.36 | 71.45 | 29.20 | 70.47 | 14.42 |
| CLIP-based + OSR | CLIP Closed Set Finetuning | 67.24 | \ | 65.19 | \ | 64.53 | \ | 66.24 | \ | 65.80 | \ |
| | CLIP Zero-Shot (Radford et al., 2021) | 52.30 | 0.81 | 50.36 | 0.26 | 46.01 | 0.38 | 47.62 | 0.25 | 49.07 | 0.43 |
| | UnivFD (Ojha et al., 2023) | 68.81 | 3.88 | 64.00 | 2.48 | 63.21 | 0.73 | 66.34 | 8.22 | 65.59 | 3.83 |
| | CLIPing (Khan & Dang-Nguyen, 2024) | 66.44 | 14.38 | 62.41 | 6.09 | 61.29 | 4.92 | 66.26 | 19.27 | 64.10 | 11.16 |
| | $D^3$ (Yang et al., 2025) | 70.46 | 8.14 | 64.71 | 8.90 | 61.65 | 1.17 | 66.33 | 8.26 | 65.79 | 6.62 |
| | **Ours** | 71.37 | **33.61** | **66.83** | **34.92** | **75.52** | **34.71** | **74.48** | **82.18** | **72.05** | **46.35** |

# 6 EXPERIMENTS

**Datasets.** To evaluate model performance on the OSFFD problem, we conducted experiments using the comprehensive dataset DF40 (Yan et al., 2024b). DF40 collects fake faces from four distinct categories ("Face Swapping", "Face Reenactment", "Entire Face Synthesis", and "Face Editing") and includes a total of 40 diverse forgeries. Additionally, we introduced data from two "Stacked Manipulation" (SM) forgeries (He et al., 2021), in which techniques from multiple fake categories are applied within a single image. We treat these SM forgeries as an auxiliary fake category.

**Evaluation Protocols.** In OSFFD problem, the training set comprises real faces and fake faces from multiple known fake categories, while the test set additionally includes samples from unknown

fake categories. To evaluate the model's performance, we first adopted the leave-one-out strategy in which one fake category from FS, FR, or EFS was withheld during training and treated as an unseen category during testing. Subsequently, all three fake categories (FS, FR, and EFS) were included as seen classes, and the model was evaluated on a test set containing additional forgeries from FE and SM, representing novel fake categories. As for the evaluation metric, we employed the multiclass classification Accuracy (**Acc**) and the Detection Rate (**DR**), where DR refers to the recall of the unseen fake categories.

We compared our DLED method with the following baseline methods. 1) Two-stage baselines: We introduced a second training stage for one-class out-of-distribution (OOD) detection methods: OC-FakeDetect (Khalid & Woo, 2020) and SBI (Shiohara & Yamasaki, 2022), in which an additional closed set multiclass model is independently trained to further classify the seen classes. For fair comparison, we used CLIP as the multiclass model's backbone and finetuned it in the closed set manner with the cross-entropy loss; 2) CNN-based baselines: Xception (Rossler et al., 2019), SPSL (Liu et al., 2021), SIA (Sun et al., 2022), UCF (Yan et al., 2023a), and NPR (Tan et al., 2024); 3) CLIP-based baselines: Zero-shot CLIP (Radford et al., 2021) and three established methods UnivFD (Ojha et al., 2023), CLIPing (Khan & Dang-Nguyen, 2024) and $D^3$ (Yang et al., 2025).

For the two-stage baselines, we let images recognized by the one-class model as seen classes go through the multiclass model to get their concrete class in testing. For the CNN-based and CLIP-based baselines, we replaced their binary classifier with a multi-class classifier trained in an end-to-end fashion and adopted the MaxLogit (Zhang & Xiang, 2023) technique in testing, because of its good performance in detecting unknown samples. For all algorithms that need a threshold to detect novel categories, we computed it from the training data such that 95% of the samples in each class are marked as known, which is the widely used setup in open set problems. Full implementation details are provided in the supplementary.

## 6.1 EVALUATION OF DETECTION PERFORMANCE

**Open Set Face Forgery Detection.**

Since the two-stage baselines rely on the closed set finetuned CLIP model as their multiclass classifier, we also report the performance of this model independently. As shown in Table 1, most baseline models struggle to achieve high performance on both Accuracy (Acc) and Detection Rate (DR) simultaneously. Methods with higher Acc typically exhibit lower DR, and vice versa. It could also be observed that two-stage methods yield lower Acc than their base forgery classifier, indicating that OOD detectors and forgery classifiers are difficult to integrate in OSFFD with satisfactory performance. Besides, directly applying OSR techniques with an Xception backbone attains notably low Acc, underscoring that off-the-

Table 2: Comparisons of prediction accuracy with diverse baselines implemented by ourselves for the Real-vs-Fake detection task. Data configurations are the same as those in OSFFD. All baseline models are implemented following their original algorithms.

| Methods | FS | FR | EFS | FE & SM | Avg |
|---|---|---|---|---|---|
| OC-FakeDect (Khalid & Woo, 2020) | 48.09 | 48.45 | 48.18 | 47.16 | 47.97 |
| SBI (Shiohara & Ya-masaki, 2022) | 50.13 | 50.36 | 50.07 | 49.96 | 50.13 |
| Xception (Rossler et al., 2019) | 71.73 | 67.98 | 67.19 | 67.49 | 68.60 |
| SPSL (Liu et al., 2021) | 72.29 | 65.87 | 70.34 | 69.57 | 69.52 |
| SIA (Sun et al., 2022) | 69.45 | 64.13 | 66.91 | 64.64 | 66.28 |
| UCF (Yan et al., 2023a) | 71.10 | 64.78 | 65.18 | 67.98 | 67.26 |
| NPR (Tan et al., 2024) | 80.76 | 75.73 | 77.67 | 77.21 | 77.84 |
| CLIP Zero-Shot (Radford et al., 2021) | 52.96 | 53.20 | 53.12 | 56.62 | 53.97 |
| UnivFD (Ojha et al., 2023) | 77.64 | 76.83 | 79.33 | 81.31 | 78.78 |
| CLIPing (Khan & Dang-Nguyen, 2024) | 78.46 | 77.15 | 79.58 | 81.09 | 79.07 |
| $D^3$ (Yang et al., 2025) | 78.56 | 77.00 | 79.67 | 79.81 | 78.76 |
| **Ours** | **87.22** | **85.93** | **83.52** | **84.97** | **85.41** |

shelf OSR approaches are insufficient to solve the OSFFD problem. With more sophisticated designs tailored to face forgery detection, the baselines achieve higher Acc in most cases, confirming that efficient mechanisms for exploring forgery-specific representations are necessary to address OSFFD.

In comparison, our DLED model consistently achieves the highest DR across all scenarios and demonstrates superior average Acc, outperforming baseline methods in the majority of cases. These results highlight the effectiveness of DLED in discovering novel fake categories while maintaining strong recognition performance on real images and known fake categories.

**Real-vs-Fake Detection.** We also evaluate the proposed DLED model on the traditional Real-vs-Fake detection task, using the same data configuration as in the OSFFD problem. In this task, all baseline methods are implemented according to their original designs without modification. For our DLED model, any face predicted to belong to a fake category is classified as a fake sample. The results are shown in Table 2. It can be observed that DLED significantly outperforms these face forgery detection algorithms across all evaluation cases. These empirical results demonstrate that, in

addition to its strong performance on the OSFFD problem, the proposed DLED model also achieves competitive results on the traditional binary Real-vs-Fake deepfake detection task.

## 6.2 Ablation Study

In this section, we conducted an ablation study on DLED. These experiments follow the same setup as described for OSFFD, and the results are summarized in Table 3.

Our results indicate that: 1) Compared to MaxLogit, EDL enhances model performance across both the spatial and frequency branches, indicating its superior capability in uncertainty estimation and, consequently, improved discovery of novel categories. 2) Although equipped with EDL, the pretrained CLIP model cannot be directly applied to the OSFFD problem in either the spatial or frequency domain, as indicated by

Table 3: Ablation Study of DLED. The table presents DR results under the same data configuration as used in the main OSFFD experiments.

| Models | | FS | FR | EFS | FE & SM | Avg |
|---|---|---|---|---|---|---|
| Spatial Branch | Zero-Shot with MaxLogit | 0.81 | 0.26 | 0.38 | 0.25 | 0.42 |
| | Zero-Shot with EDL | 1.58 | 0.58 | 0.68 | 0.63 | 0.87 |
| | Finetuning with EDL | 13.02 | 30.94 | 8.33 | 50.59 | 25.71 |
| Frequency Branch | Zero-Shot with MaxLogit | 3.85 | 2.35 | 6.98 | 0.53 | 3.43 |
| | Zero-Shot with EDL | 4.71 | 2.51 | 6.06 | 0.55 | 3.46 |
| | Finetuning with EDL | 14.34 | 8.49 | 7.69 | **90.36** | 30.22 |
| Two Branches | Evidence Fusion | 32.42 | **36.16** | 32.56 | 79.74 | 45.22 |
| | Full DLED | **33.61** | 34.92 | **34.71** | 82.18 | **46.36** |

its extremely poor performance (see the 2nd and 5th rows). Fine-tuning the prompts and integrating LoRA layers substantially improves the performance of both branches, highlighting the effectiveness of task-specific representation adaptation. 3) Without frequency information, the finetuned spatial branch with EDL exhibits an average performance drop of about 20% relative to the fused model (see the 3rd and 7th rows). This highlights the necessity of extracting complementary evidential cues across spatial and frequency domains to fully exploit forgery-specific signals and make more effective use of EDL, as well as the benefits of evidence integration. 4) By incorporating the improved uncertainty estimation, the full DLED model achieves the highest average Detection Rate, surpassing simple evidence fusion in most cases and thereby validating its effectiveness.

## 6.3 Analysis of Evidence

To provide a clearer understanding of DLED's behavior in OSFFD, we present visualizations of evidence distribution in Fig. 4. In this analysis, FR and EFS are treated as seen fake categories, while FS and FE represent novel categories.

Fig. 4 illustrates how uncertainty estimation facilitates the detection of novel fake categories among test samples. Each subfigure visualizes the Dirichlet distribution produced by DLED for the corresponding fake category. These visualizations demonstrate that the DLED model exhibits higher confidence when making predictions on seen classes, while showing greater prediction uncertainty for novel fake categories. This behavior enables DLED to effectively recognize newly emerging fake categories while simultaneously maintaining strong performance on known classes.

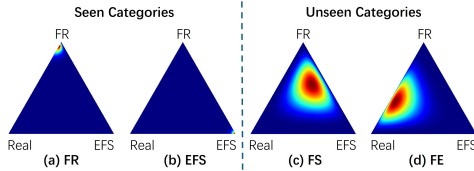

Figure 4: **Visualization of Evidence Distribution.** The evidence for seen fake categories FR and EFS is condensed in their corresponding corner with low uncertainty, while the evidence for novel fake categories FS and FE is sparse with higher uncertainty.

## 7 Conclusion

In this work, we reformulate the Open Set Face Forgery Detection (OSFFD) problem by removing the need for unlabeled novel data during model training, thereby enhancing its practicality for real-world applications. By treating the OSFFD as an uncertainty estimation problem, we proposed a novel algorithm, DLED, which effectively identifies unseen fake categories as novel while simultaneously classifying real and known fake categories. DLED leverages EDL to collect and fuse evidence from both spatial and frequency domains, exploiting category-specific semantics to estimate prediction uncertainty. Additionally, we propose an improved uncertainty formulation that enhances the model's ability to detect novel fake categories. Extensive experiments under various testing configurations demonstrate that DLED substantially outperforms diverse baseline methods in addressing the OSFFD problem. Future work will focus on improving the efficiency of the proposed method and enabling rapid adaptation to the detected novel fake categories.

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
