# Open Set Face Forgery Detection via Dual-Level Evidence Collection (Supplementary Materials)

## A  Dataset Setup Details

Table 1: Forgery configuration of each fake category in our OSFFD setting. This table details the specific deepfake forgeries included in each fake category. Certain forgeries are used in both the training and testing phases, while others are reserved exclusively for testing.

|  | FS | FR | EFS | FE | SM |
|---|---|---|---|---|---|
| Train & Test | FSGAN, FaceSwap, SimSwap, InSwapper, BlendFace | FOMM, FS_vid2vid, Wav2Lip, MRAA, OneShot | VQGAN, HStyleGAN-XL, SD-2.1, StyleGAN2, StyleGAN3 | \ | \ |
| Test Only | UniFace, MobileSwap, e4s, FaceDancer, DeepFaceLab | LIA, TPSMM, HyperReenact, DaGAN, SadTalker, MCNet, PIRender, HeyGen | PixArt-alpha, MinJourney6, DDPM, WhichisReal, RDDM, DiT-XL/2, SiT-XL/2 | StarGAN, StarGANv2, e4e, CollabDiff, StyleCLIP | DeepFakes-StarGan-Stack, StarGan-BlendFace-Stack |

Here, we provide the data configuration details about our setup in our Open Set Face Forgery Detection (OSFFD) experiments. We utilize the DF40 (Yan et al., 2024) dataset, which comprises four fake categories, and further incorporate two Stacked Manipulation forgeries from the ForgeryNet benchmark (He et al., 2021), resulting in five distinct fake categories in total. Specifically, the five fake categories are: Face Swapping (FS), Face Reenactment (FR), Entire Face Synthesis (EFS), Face Editing (FE), and Stacked Manipulation (SM). Due to the limited amount of fake data available for FE and SM, these two categories are treated as novel fake categories. The remaining categories FS, FR, and EFS are considered as either seen or unseen categories across different experimental configurations. As outlined in the main paper, we initially adopted a leave-one-out strategy, in which one of the three categories was withheld during training and regarded as an unseen category during testing. In a subsequent setting, all three categories (FS, FR, and EFS) were included as known classes, while FE and SM were used as novel fake categories during evaluation. During the training phase, the data consists of faces form real class and known fake categories, whereas the testing phase additionally includes samples from novel fake categories.

Since the number of forgeries varies across the fake categories FS, FR, and EFS, we select five forgeries from each category for use in both training and testing, while reserving the remaining forgeries for testing only. We show the forgery configuration in Table 1. For detailed information on the methods included in FS, FR, EFS, and FE, along with the corresponding data quantities, sources, and sub-types, please refer to Table 2 in the DF40 paper (Yan et al., 2024). The methods categorized under SM can be found in Figure 3 of the ForgeryNet benchmark paper (He et al., 2021). We followed the original training and testing splits for each specific deepfake forgery and performed dataset balancing by randomly selecting the largest possible subset such that each fake category contained an equal number of samples, and the overall number of real and fake face images was balanced.

## B  Implementation Details

**Network Modules.** Our Dual-Level Evidential face forgery Detection (DLED) method employs CLIP (Radford et al., 2021) as the backbone for both the spatial and frequency branches, using the ViT-B/16 (Dosovitskiy, 2020) encoder with half-precision (fp16) data type. The same CLIP configuration is also used for all CLIP-based baseline methods to ensure fair comparisons. For the CLIP Zero-Shot baseline, we used the text prompt "a [CLASS] photo." as the input, where "[CLASS]" is replaced with the predefined class names corresponding to the seen classes. For CNN-based baseline methods and one-class detectors in Two-stage baselines, all backbones utilize the official implementations provided by the respective authors. These baseline models are trained

from scratch for the OSFFD experiments. Specifically, SPSL (Liu et al., 2021) and UCF (Yan et al., 2023) utilized the Xception backbone (Rossler et al., 2019), while SBI (Shiohara & Yamasaki, 2022) and SIA (Sun et al., 2022) are based on EfficientNet-B4 (Tan & Le, 2019). NPR (Tan et al., 2024) is a modified version of ResNet (He et al., 2016), and OC-FakeDetect (Khalid & Woo, 2020) incorporates a Variational Autoencoder. (Kingma & Welling, 2013). Xception (Rossler et al., 2019) is used directly without modification.

**Concrete Implementation.** OSFFD requires the model to simultaneously detect novel fake categories and perform multiclass classification. Accordingly, we use multiclass classification accuracy (Acc) to evaluate performance on classification, and Detection Rate (DR), defined as the recall of the unseen fake categories, to assess the model's ability to discover novel fake. In our experiments, the batch size was set to 32 for training and 100 for testing across all models. Our DLED model employed the softplus function as its confidence function $h(\cdot)$. Both DLED and CLIP-based baseline models are optimized using SGD with a learning rate of 0.002, while the remaining baseline methods use the Adam optimizer with a learning rate of $1e^{-4}$. All models, except for the CLIP Zero-Shot baseline, are trained for 50 epochs. We randomly selected a seed and evaluated the model's performance at the final epoch in a single run. All methods are implemented in PyTorch, and experiments are conducted on an RTX 6000 Ada GPU. Our code is available at https://anonymous.4open.science/r/NovelDFD-BF78.

**AI Disclosure.** We used generative AI to edit portions of this paper for clarity, grammar, and word choice.

## C  DLED ALGORITHM

In this section, we provide a detailed description of the inference procedure for our DLED model. We denote the spatial and frequency branches as $\mathcal{F}^s$ and $\mathcal{F}^f$, respectively, such that $\mathcal{F} = \{\mathcal{F}^s, \mathcal{F}^f\}$. Given the retrieved uncertainty thresholds $\{\tau_k\}_{k=1}^K$ for each known class $k$, the DLED inference procedure for an input face image $x$ is described in Algorithm 1, where $h(\cdot)$ denotes the evidence function and $\mathrm{FFT}(\cdot)$ represents the Fast Fourier Transform.

---

**Algorithm 1** DLED Inference Procedure

---

**Require:** Input image $x$; uncertainty thresholds $\{\tau_k\}_{k=1}^K$ for each known class.

1: Obtain the frequency map: $x^f = \mathrm{FFT}(x)$
2: Extract dual-level evidence: $e^s = h(\mathcal{F}^s(x))$, $e^f = h(\mathcal{F}^f(x^f))$
3: Calculate belief and uncertainty for each branch:
$\quad b_k = e_k/S, \quad u = K/S, \quad S = \sum_{k=1}^K (e_k + 1)$
4: Fuse belief and uncertainty:
$\quad \widetilde{b}_k = \gamma(b_k^s b_k^f + b_k^s u^f + b_k^f u^s),$
$\quad \widetilde{u} = \gamma u^s u^f, \quad \gamma = 1/\left(1 - \sum_{i \neq j} b_i^s b_j^f\right)$          [Eq. 5]
5: Calculate fused evidence and parameters:
$\quad \widetilde{S} = K/\widetilde{u}, \quad \widetilde{e}_k = \widetilde{b}_k \times \widetilde{S}, \quad \widetilde{\alpha}_k = \widetilde{e}_k + 1$          [Eq. 6]
6: Calculate predictive distribution and predicted label:
$\quad \widetilde{p}_k = \widetilde{\alpha}_k/\widetilde{S}, \quad \widetilde{y} = \arg\max_k \widetilde{p}_k$
7: Improve uncertainty estimation:
$\quad \hat{u} = 1/\left(\max\{\widetilde{\alpha}_1, \ldots, \widetilde{\alpha}_K\}\right)$          [Eq. 8]
8: Novel category detection and final prediction:
$\quad \hat{y} = \begin{cases} \widetilde{y}, & \text{if } \hat{u} \leq \tau_{\widetilde{y}} \\ K+1, & \text{if } \hat{u} > \tau_{\widetilde{y}} \end{cases}$
9: **return** $\hat{y}$ and $\hat{u}$

---

## D  INFLUENCE OF UNCERTAINTY THRESHOLD

In this section, we present additional experimental results analyzing the impact of the uncertainty threshold on the OSFFD problem. Uncertainty threshold is calculated from the training data such

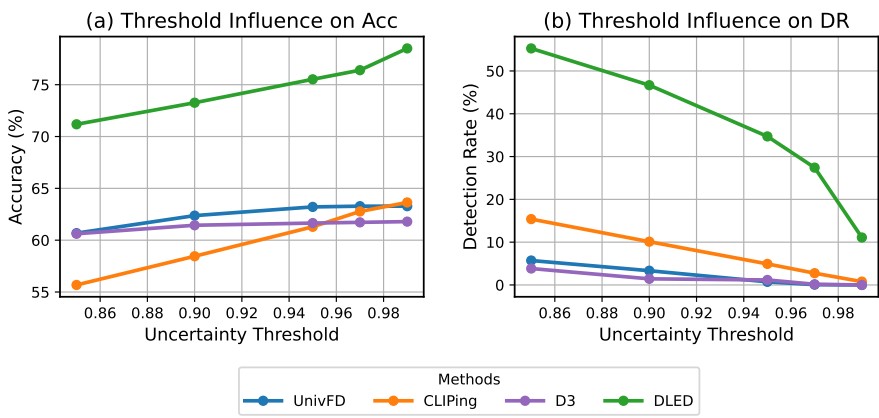

Figure 1: Accuracy and Detection Rate across different uncertainty thresholds for UnivFD, CLIPing, $D^3$, and DLED.

that a certain ratio of the samples in each class are marked as known. Our experiments on the OSFFD problem followed the widely adopted open set recognition evaluation protocol (Wang et al., 2022; Zhao et al., 2023), which sets the uncertainty threshold at $95\%$. A higher threshold indicates that more training samples are regarded as reliable, thereby imposing a stricter criterion for novel fake discovery. To further investigate the impact of the uncertainty threshold on the OSFFD problem, we conducted a series of experiments using various threshold values [0.85, 0.90, 0.95, 0.97, 0.99] on DLED, as well as three CLIP-based baseline methods: UnivFD (Ojha et al., 2023), CLIPing (Khan & Dang-Nguyen, 2024), and $D^3$ (Yang et al., 2025).

In this experiment, FS and FR are treated as seen fake categories, while EFS is designated as the only novel fake category. The results are presented in Figure 1. It can be observed that as the uncertainty threshold increases, classification accuracy improves, whereas the novel fake discovery rate declines. This indicates that a stricter threshold reduces the detection of novel samples but also decreases the misclassification of known samples. These findings highlight the critical role of the uncertainty threshold in balancing known class classification and novel fake discovery. Moreover, across all threshold settings, our DLED method consistently achieves higher Acc and DR compared to the three CLIP-based baselines, demonstrating its robustness and superior performance.