# OpenReview forum: "Open Set Face Forgery Detection via Dual-Level Evidence Collection"
_ICLR.cc/2026/Conference — ICLR 2026 Conference Withdrawn Submission_

### Official Review · Reviewer_1CK1 · 2025-10-24

**Soundness:** 3
**Presentation:** 3
**Contribution:** 3
**Rating:** 4
**Confidence:** 4

**Summary:**

This paper addresses the Open Set Face Forgery Detection (OSFFD) problem, which requires a model to not only classify forgeries from known categories but also to identify forgeries generated by novel, previously unseen methods. The authors first reformulate the OSFFD problem to enhance its practical relevance by removing the need for unlabeled data from novel categories during training; the model is trained exclusively on real images and known forgery types.1
To tackle this reformulated problem, the paper introduces the Dual-Level Evidential face forgery Detection (DLED) method. DLED frames OSFFD as an uncertainty estimation task, leveraging Evidential Deep Learning (EDL) to quantify model confidence. The core of the DLED architecture is a dual-branch CLIP-based model that collects evidence from two complementary domains: high-level spatial semantics from the raw image and low-level artifacts from its frequency-domain representation.1 At test time, evidence from these two branches is fused using Dempster's combination rule to produce a unified belief and uncertainty score. The authors also propose an alternative uncertainty metric, which uses the maximum evidence across classes instead of the average, to improve the detection of novel categories.
Experiments are conducted on the comprehensive DF40 dataset. The results demonstrate that DLED significantly outperforms a wide range of baselines in detecting novel forgery categories, achieving a much higher Detection Rate (DR) while maintaining competitive multiclass classification accuracy (Acc). The method also shows strong performance on the traditional binary Real-vs-Fake detection task.

**Strengths:**

(S1) Timely and Well-Motivated Problem Formulation: The paper tackles a critical and forward-looking problem in digital media forensics. The reformulation of OSFFD is a significant strength. By removing the dependency on unlabeled novel-category data during training, the proposed setting is far more aligned with real-world scenarios where new forgery methods emerge without warning.1 This makes the research more impactful and applicable. Furthermore, the paper does an excellent job of distinguishing OSFFD from standard deepfake detection (binary), deepfake attribution (closed-set multiclass), and simple OOD generalization, providing a clear context for its unique contributions (1, Figure 1).
(S2) Novel and Principled Methodological Approach: The DLED architecture is well-designed and theoretically grounded. The fusion of spatial and frequency domain information is a well-established concept in forgery detection, but its integration within an EDL framework to collect complementary "evidence" is a novel and powerful insight.1 This approach is well-supported by the ablation study in Table 3, which convincingly demonstrates that both branches are necessary and that the fused model significantly outperforms either branch alone. Framing the problem in terms of uncertainty via EDL is an elegant fit for open-set recognition, moving beyond simple post-hoc scoring functions (like MaxLogit) to provide a more principled way to quantify "known unknowns".1 The use of Dempster's rule for fusion is also a theoretically sound choice for combining evidence from independent sources, as demonstrated in related multi-view EDL applications.
(S3) Comprehensive and Rigorous Evaluation: The experimental validation is thorough and largely convincing. The authors compare DLED against a wide array of relevant baselines, including two-stage OOD detectors, CNN-based forgery detectors, and recent CLIP-based methods, providing a strong context for evaluating its performance.1 The choice of both multiclass Accuracy (Acc) and novel-category Detection Rate (DR) is crucial, as it correctly captures the dual objectives of the OSFFD task. The results in Table 1 clearly show DLED striking a much better balance between these two metrics than the baselines. The ablation study in Table 3 is exemplary, as it systematically dissects the DLED model to validate the contribution of each key component: the use of EDL over MaxLogit, the necessity of fine-tuning CLIP, the benefit of the frequency branch, and the impact of the improved uncertainty estimation. This analysis greatly strengthens the authors' claims.
(S4) High-Quality Presentation: The paper is well-written, clearly structured, and easy to follow. The figures, particularly Figure 1 and Figure 3, are informative and effectively communicate the core ideas. The mathematical formulations are generally clear and self-contained.

**Weaknesses:**

(W1) Overstated Novelty and Insufficient Justification of the "Improved Uncertainty Estimation": The paper presents the proposed uncertainty metric as a key contribution, but its novelty and superiority over other EDL formulations are not well-established. The proposed metric replaces the average evidence in the denominator of the standard uncertainty calculation with the maximum evidence, yielding an expression of the form $\hat{u} = 1 / \max\{\tilde{\alpha}_{1,...,K}\}$.1 While the authors correctly identify a potential issue with the standard EDL uncertainty—that high evidence from irrelevant classes can artificially lower the uncertainty score—their solution is conceptually very similar to post-hoc OOD detection methods like Maximum Softmax Probability or MaxLogit, which also rely on the confidence of the top prediction. The paper itself introduces MaxLogit as a baseline OSR technique, which raises the question of whether this is a truly "improved" formulation of evidential uncertainty or simply a MaxLogit-style heuristic applied within the EDL framework.
The work does not provide a theoretical justification for why this is a more principled uncertainty measure under Dempster-Shafer theory. Recent research has explored more fundamental ways to improve EDL, such as rethinking the loss function or adjusting prior weights, suggesting that the original formulation has deeper issues that may not be solved by such a heuristic (Deng et al., 2024). The ablation study shows a performance gain, but it is not clear if this is due to a fundamentally better uncertainty representation or a better-tuned heuristic. A more convincing analysis would compare this proposed metric against other simple uncertainty scores derived from the same evidence vector (e.g., entropy of the belief masses).
(W2) Baselines for Open-Set Recognition Could Be Stronger and More Modern: While the set of forgery detection baselines is strong, their adaptation to the open-set problem via MaxLogit may not represent the state-of-the-art, especially for powerful CLIP/ViT-based models.1 MaxLogit is a simple, post-hoc method that relies solely on the network's final output logits. However, a growing body of work on OOD detection for Transformer architectures has shown that more sophisticated methods leveraging intermediate feature representations are often superior. For example, methods like BLOOD (Jelenic et al., 2024) analyze the smoothness of transformations between intermediate layers to detect OOD inputs and have shown strong performance on Transformer-based models. By only using MaxLogit, the paper might be under-representing the true capability of these powerful CLIP-based backbones for the OSFFD task. The best-performing OOD methods can be highly architecture-dependent, a nuance that is missing from the current paper's evaluation, which applies a one-size-fits-all adaptation to all baselines.
(W3) Lack of Efficiency Analysis: The proposed DLED model uses two parallel CLIP pipelines, which implies a significant computational cost. The paper completely omits any discussion of efficiency, inference latency, or computational complexity (e.g., FLOPs). The model overview in Figure 3 clearly shows two separate branches, and while parameters are shared, this still requires two forward passes through the vision encoder.1 Deepfake detection is often a real-time application, making inference speed a critical practical consideration. The baselines are mostly single-stream models, so a fair comparison must also consider the performance-per-compute trade-off. Recent work in deepfake detection has increasingly emphasized efficiency, with some methods specifically designed for real-time performance on resource-constrained devices (Lanzino et al., 2024). The current paper is positioned at the high-performance, high-cost end of the spectrum without acknowledging or justifying this trade-off.

References Cited in This Review
Deng, Z., Liu, T., Zhou, Z., & Liu, T. (2024). Revisiting Essential and Nonessential Settings of Evidential Deep Learning. arXiv preprint arXiv:2410.00393.
Jelenic, F., Jukic, J., Tutek, M., Puljiz, M., & Šnajder, J. (2024). Out-of-Distribution Detection by Leveraging Between-Layer Transformation Smoothness. In International Conference on Learning Representations (ICLR).
Lanzino, R., Fontana, F., Diko, A., Marini, M. R., & Cinque, L. (2024). Faster Than Lies: Real-time Deepfake Detection using Binary Neural Networks. In Proceedings of the IEEE/CVF Conference on Computer Vision and Pattern Recognition (CVPR) Workshops (pp. 3771-3780).

**Questions:**

See weakness

---

### Official Review · Reviewer_Fw7w · 2025-10-28

**Soundness:** 2
**Presentation:** 3
**Contribution:** 3
**Rating:** 4
**Confidence:** 5

**Summary:**

This paper addresses the Open Set Face Forgery Detection (OSFFD) problem. The authors note that existing face forgery detection methods are typically limited to binary real-vs-fake classification or the identification of known fake categories and are unable to detect novel types of forgeries. To overcome this, they reformulate OSFFD as an uncertainty estimation problem and propose a Dual-Level Evidential face forgery Detection (DLED) method. DLED collects and fuses category-specific evidence in both the spatial and frequency domains to estimate prediction uncertainty. Extensive evaluations across a variety of experimental settings demonstrate that DLED achieves state-of-the-art performance in detecting forgeries from novel fake categories and remains competitive on traditional real-versus-fake face forgery detection tasks.

**Strengths:**

- This manuscript is well-structured and easy to follow.

**Weaknesses:**

- The uncertainty-estimation-based fusion strategy has already been discussed in [1] for deepfake detection, but the manuscript neither cites this work nor provides a comparison.

- The analysis of uncertainty estimation is not sufficiently in-depth. As highlighted in the recent survey on EDL [2], Dirichlet-based uncertainty typically covers four representative scenarios, whereas the paper only presents the ideal “DC” (dominant & certain) case. To make the discussion more complete, the remaining three scenarios should also be analyzed and, preferably, illustrated with qualitative examples tailored to the deepfake detection task.

[1] Face Forgery Detection with Elaborate Backbone. arXiv: 2409.16945.

[2] A Comprehensive Survey on Evidential Deep  Learning and Its Applications. TPAMI 2025.

**Questions:**

- Does it really make sense to align the frequency-domain encoder with the text domain? CLIP-style contrastive learning is designed for semantic alignment between images and text, assuming the visual features carry explicit, human-interpretable semantics that can be grounded in language. However, frequency maps (FFT-based inputs) mainly encode artifact-level or statistical cues rather than object/scene semantics. In this case, forcing the frequency branch to share the same image–text alignment space may not be theoretically justified. It would be helpful if the authors could clarify the motivation for using CLIP-like alignment on frequency features.

- Although the authors provide a code link in the supplementary material, it currently returns “The requested file is not found.”

**Details Of Ethics Concerns:**

No ethical concerns identified.

---

### Official Review · Reviewer_gZzC · 2025-10-29

**Soundness:** 2
**Presentation:** 2
**Contribution:** 2
**Rating:** 2
**Confidence:** 5

**Summary:**

This work addresses the problem of open-set face forgery detection, which focuses on identifying previously unseen forgery types. The proposed OSFFD framework introduces a dual-level evidential detection architecture that integrates both spatial and frequency-domain features. Experimental results demonstrate that the method achieves good detection performance on novel forgery types.

**Strengths:**

The paper is well organized and easy to follow. The proposed method outperforms previous approaches on DF40 and FF++ datasets.

**Weaknesses:**

- The significance of this work appears limited. Previous studies have extensively explored the generalizability of deepfake detectors against unseen forgeries and achieved impressive results. Moreover, existing deepfake attribution methods can already classify forgery types with high accuracy. The practical application and necessity of the proposed task remain unclear, rendering the overall contribution incremental.

- The novelty of the proposed approach is limited. The integration of spatial and frequency branches has been widely investigated in earlier deepfake detection literature. In addition, the adopted evidential uncertainty estimation technique was originally proposed in 2018 and has since been utilized in various related works.

- The experimental evaluation is insufficiently comprehensive. The method is only tested on the DF40 and FF++ datasets. To convincingly demonstrate its superiority and robustness, the authors should consider including experiments on more challenging and diverse benchmark datasets.

- The manuscript lacks comparisons with sota methods. Even if this is among the first works addressing open-set face forgery localization, it is essential to include performance comparisons with recent deepfake detection methods published in top-tier conferences and journals to better contextualize the contribution.

- Some inaccurate claims are present in the manuscript. For example, on page 6, the authors state that most existing works neglect high-level spatial semantic signals and low-level frequency artifacts. In fact, these characteristics have been thoroughly investigated in numerous previous deepfake detection studies.

**Questions:**

NA

---

### Official Review · Reviewer_eLPe · 2025-10-31

**Soundness:** 2
**Presentation:** 3
**Contribution:** 2
**Rating:** 4
**Confidence:** 5

**Summary:**

This paper reformulated the OSFFD problem and address it through uncertainty estimation, enhancing its applicability to real-world scenarios. Experiments show that the proposed model achieves state-of-the-art performance.

**Strengths:**

Strengths:
- The motivation of this paper is clear and the authors chose a straightforward but effective method to achieve the goal.
- The charts related to the experiments in the paper are relatively clear, and the organization of the charts is logical.

**Weaknesses:**

Weaknesses:
- The paper directly applies EDL to uncertainty estimation in OSFFD, but fails to analyze the differences between the feature distribution of forged face images and the applicable scenarios of EDL. For example, does EDL produce excessive uncertainty estimation when dealing with forged samples with low evidence strength? This issue is not discussed, and the application of EDL lacks specific justification.
- The paper mentions integrating a LoRA layer into the CLIP encoder, but it doesn't specify key parameters such as the LoRA's rank and fine-tuning learning rate, nor does it verify the impact of different parameters on model performance. The appropriateness of parameter selection may affect model efficiency and performance; it is recommended to supplement relevant details and ablation experiments.
- The paper mentions using an evidence function h(⋅) to give the predicted evidence, but it does not specify the exact form of this function or the basis for its selection. Different evidence functions may affect the rationality of evidence quantification; it is recommended to supplement the function definition and comparative experiments to verify its impact on the results.
- DLED employs a dual CLIP branch structure. While it mentions " reduce memory consumption by sharing their pretrained parameters," it does not provide specific data on the number of parameters, FLOPs, or inference time. Compared to lightweight baselines (such as Xception), the computational cost of the dual CLIP structure is significantly higher, and the paper fails to demonstrate whether its performance improvement is sufficient to offset the efficiency loss, thus failing to meet the needs of real-world scenarios such as real-time detection.

**Questions:**

- Improvements to the baseline method lacked consistency. For example, in the “CNN-based + OSR” baseline, the binary classifier was replaced with a multi-classifier, but it was not explained whether this modification was consistent with the original method's design intent. This could lead to an underestimation of the baseline performance and unfair comparison results.
- The experiments were conducted solely on the DF40 dataset, and the novel forgery categories were limited to Face Editing (FE) and Stacked Manipulation (SM), failing to cover mainstream novel forgery techniques in recent years (such as face generation based on new AR models). Furthermore, complex scene interference was not considered, severely limiting the generalization ability of the experimental results and making it impossible to prove the effectiveness of the method in the real world.

---

### Note · Authors · 2025-12-31

I have read and agree with the venue's withdrawal policy on behalf of myself and my co-authors.